# The Arthropoda-specific Tramtrack group BTB protein domains use previously unknown interface to form hexamers

Artem N Bonchuk[1,2]*, Konstantin I Balagurov[1], Rozbeh Baradaran[3], Konstantin M Boyko[4], Nikolai N Sluchanko[4], Anastasia M Khrustaleva[1], Anna D Burtseva[4,5], Olga V Arkova[2], Karina K Khalisova[1], Vladimir O Popov[4], Andreas Naschberger[3]*, Pavel G Georgiev[1]*

[1]Department of the Control of Genetic Processes, Institute of Gene Biology Russian Academy of Sciences, Moscow, Russian Federation; [2]Center for Precision Genome Editing and Genetic Technologies for Biomedicine, Institute of Gene Biology, Russian Academy of Sciences, Moscow, Russian Federation; [3]Bioscience Program, Division of Biological and Environmental Science and Engineering (BESE), King Abdullah University of Science and Technology (KAUST), Thuwal, Saudi Arabia; [4]Bach Institute of Biochemistry, Research Center of Biotechnology Russian Academy of Sciences, Moscow, Russian Federation; [5]Moscow Institute of Physics and Technology, Institutsky lane 9, Moscow, Russian Federation

*For correspondence:
echinaceus@gmail.com (ANB);
andreas.naschberger@kaust.edu.
sa (AN);
georgiev_p@mail.ru (PGG)

Competing interest: The authors declare that no competing interests exist.

**Abstract** BTB (bric-a-brack, Tramtrack, and broad complex) is a diverse group of protein-protein interaction domains found within metazoan proteins. Transcription factors contain a dimerizing BTB subtype with a characteristic N-terminal extension. The Tramtrack group (TTK) is a distinct type of BTB domain, which can multimerize. Single-particle cryo-EM microscopy revealed that the TTK-type BTB domains assemble into a hexameric structure consisting of three canonical BTB dimers connected through a previously uncharacterized interface. We demonstrated that the TTK-type BTB domains are found only in Arthropods and have undergone lineage-specific expansion in modern insects. The *Drosophila* genome encodes 24 transcription factors with TTK-type BTB domains, whereas only four have non-TTK-type BTB domains. Yeast two-hybrid analysis revealed that the TTK-type BTB domains have an unusually broad potential for heteromeric associations presumably through a dimer-dimer interaction interface. Thus, the TTK-type BTB domains are a structurally and functionally distinct group of protein domains specific to Arthropodan transcription factors.

## Editor's evaluation

This important study investigates Tramtrack–like BTB domains of metazoan transcription factors using Cryo–EM microscopy, evolutionary and fold prediction analyses. The research presents compelling evidence for the structural basis of the multimerization and explores the evolutionary history of this family. This study will be of particular interest to structural and evolutionary biologists.

## Introduction

BTB, also known as POZ (Pox-virus and zinc-finger), is an evolutionarily conserved domain that was originally found in the *Drosophila* proteins bric-a-brac, Tramtrack, and broad complex (*Zollman et al., 1994*). BTB proteins have been identified in poxviruses and eukaryotes, and have various functions including regulation of transcription, chromatin remodeling, cytoskeletal function, ion transport,

and ubiquitination/degradation of proteins (*Perez-Torrado et al., 2006*; *Stogios et al., 2005*). BTB domains are 100–120 aa in size and the core structure consists of five α-helices and three β-strands (*Ahmad et al., 1998*; *Bonchuk et al., 2023*; *Li et al., 1999*). In addition to this core, different subclasses of BTB proteins include N- and C-terminal BTB extension regions that facilitate protein-specific functions. As a result, the BTB fold is a versatile scaffold that participates in a variety of family-specific protein-protein interactions (*Bonchuk et al., 2023*).

In metazoans, BTB domains of transcription factors (commonly referred to as BTB and Zinc Fingers or ZBTB since they are mostly found in combination with C2H2 zinc fingers) contain an amino-terminal extension that enables homodimerization (*Ahmad et al., 1998*; *Ahmad et al., 2003*; *Bonchuk et al., 2023*). Other subtypes of BTB domains are Cullin-interacting KLHL-BTBs, which mostly form dimers; Skp1/ElonginC proteins which are subunits of Ubiquitin-ligase complexes; tetrameric T1 BTB domains participating in ion channel formation and the related pentameric KCTD BTB domains with various functions (*Bonchuk et al., 2023*). Most members of the ZBTB family contain C2H2-type zinc fingers (BTB-C2H2). In humans, 156 genes are predicted to encode BTB domain-containing proteins, of which 49 are transcription factors possessing between 2 and 14 C2H2 domains (*Siggs and Beutler, 2012*). In *Drosophila*, 56 genes are predicted to encode BTB proteins, out of which 28 are transcription factors with the BTB domain of the ZBTB subtype (see below). BTB-C2H2 proteins act as classical transcription factors, binding to chromatin and participating in the regulation of transcription. In insects, some transcription factors contain BTB domains in combination with other DNA-binding domains such as helix-turn-helix (HTH) and its subtype Pipsqueak (PSQ, BTB PSQ *Siegmund and Lehmann, 2002*). Several B T B-containing transcription factors contain FLYWCH (named after characteristic sequence motif) domains that belong to the WRKY family of transcription factors (*Babu et al., 2006*) and may be involved in the interaction with either DNA, RNA, or proteins (*Beaster-Jones and Okkema, 2004*; *Melnikova et al., 2017*).

In mammals, BTB-C2H2 transcription factors are required for the development of lymphocytes, fertility, skeletal morphogenesis, and neurological development (*Chaharbakhshi and Jemc, 2016*). The well-characterized BTB domains form tightly intertwined dimers and possess a peptide-binding groove, which is responsible for the interaction with various transcription factors and co-repressor complexes (*Ahmad et al., 2003*; *Ghetu et al., 2008*; *Vogelmann et al., 2014*; *Zacharchenko and Wright, 2021*). In most cases, mammalian C2H2 transcription factors have BTB domains which exclusively form homodimers. An exception to this rule is the C2H2 protein Miz-1, whose BTB domain forms tetramers (*Stead et al., 2007*). Several BTB domains are capable of heterodimer formation (*Olivieri et al., 2021*). The structural basis for heterodimerization has been studied using chimeric Bcl6/Miz-1 and Miz-1/NAC1 assembly *Stead and Wright, 2014*; however, there is evidence that formation of such heterodimers in vivo is prevented by co-translational dimer assembly and the quality-control protein degradation machinery (*Bertolini et al., 2021*; *Mena et al., 2020*; *Mena et al., 2018*).

In *Drosophila*, a number of C2H2 proteins with a BTB domain have been described, some containing a typical BTB domain, such as CP190 and CG6792 (*Maeng et al., 2012*). The most well-studied of these, CP190, has four C2H2 zinc finger domains that appear to be involved in protein-protein interactions rather than DNA binding (*Oliver et al., 2010*). The N-terminal BTB/POZ domain of CP190 forms stable homodimers (*Bonchuk et al., 2011*; *Oliver et al., 2010*; *Plevock et al., 2015*; *Sabirov et al., 2021b*; *Vogelmann et al., 2014*). CP190 is required for the activity of housekeeping promoters and insulators (*Bartkuhn et al., 2009*; *Sabirov et al., 2021a*). CG6792 is similar to mammalian BTB-C2H2 proteins, has seven DNA-binding zinc fingers, and is involved in wing development (*Maeng et al., 2012*).

Most of the well-characterized *Drosophila* BTB transcription factors, including GAF, Mod(mdg4), LOLA, Broad-complex (BR-C), Batman, Pipsqueak, and Bric-a-brac (Bab), have TTK-type BTB domains (*Bonchuk et al., 2011*; *Zollman et al., 1994*). Proteins from this group often have important functions in transcription regulation, development, and chromosome architecture (*Chaharbakhshi and Jemc, 2016*). Many TTK-group proteins, such as BR-C, TTK, and Bab, are critical regulators of development that function as transcriptional repressors (*Bradley and Andrew, 2001*; *Chaharbakhshi and Jemc, 2016*; *Mukai et al., 2007*; *Silva et al., 2016*). Several have been implicated in chromatin architectural function, acting as a component of a chromatin insulator complex (Mod(mdg4)) or recruiting chromatin remodeling complexes (GAF, Pipsqueak) (*Huang et al., 2002*; *Lomaev et al., 2017*). Recently,

it was shown that GAF is a pioneer factor that acts as a stable mitotic bookmarker during zygotic genome activation during *Drosophila* embryogenesis (*Bellec et al., 2022*; *Tang et al., 2022*).

TTK-type BTB domains contain a highly conserved N-terminal FxLRWN motif, where x is a hydrophilic residue (*Bonchuk et al., 2011*). Several TTK-type BTB domains can selectively interact with each other and form multimers (*Bonchuk et al., 2011*). Although there are more than 30 crystal structures of non-TTK BTB domains that form stable homodimers (*Ahmad et al., 1998*; *Stogios et al., 2007*; *Stogios et al., 2010*; *Stogios et al., 2005*; *Vogelmann et al., 2014*), the structures of the TTK-type BTB domains and the structural basis for their multimerization remain unknown.

In this study, we investigated the presence of TTK-type and canonical BTB domains in transcription factors from various phylogenetic groups of Bilateria. The TTK-type BTB domains were found only in Arthropodan transcription factors and underwent a lineage-specific expansion in insects. Using an integrative structural biology approach, we built a structural model of the TTK-group BTB hexameric assembly, validating it using MALS, SAXS, cryo-EM microscopy, and site-directed mutagenesis. Finally, we found an unusual potential for these domains to form heteromultimers, which likely involve a dimer-dimer interaction interface.

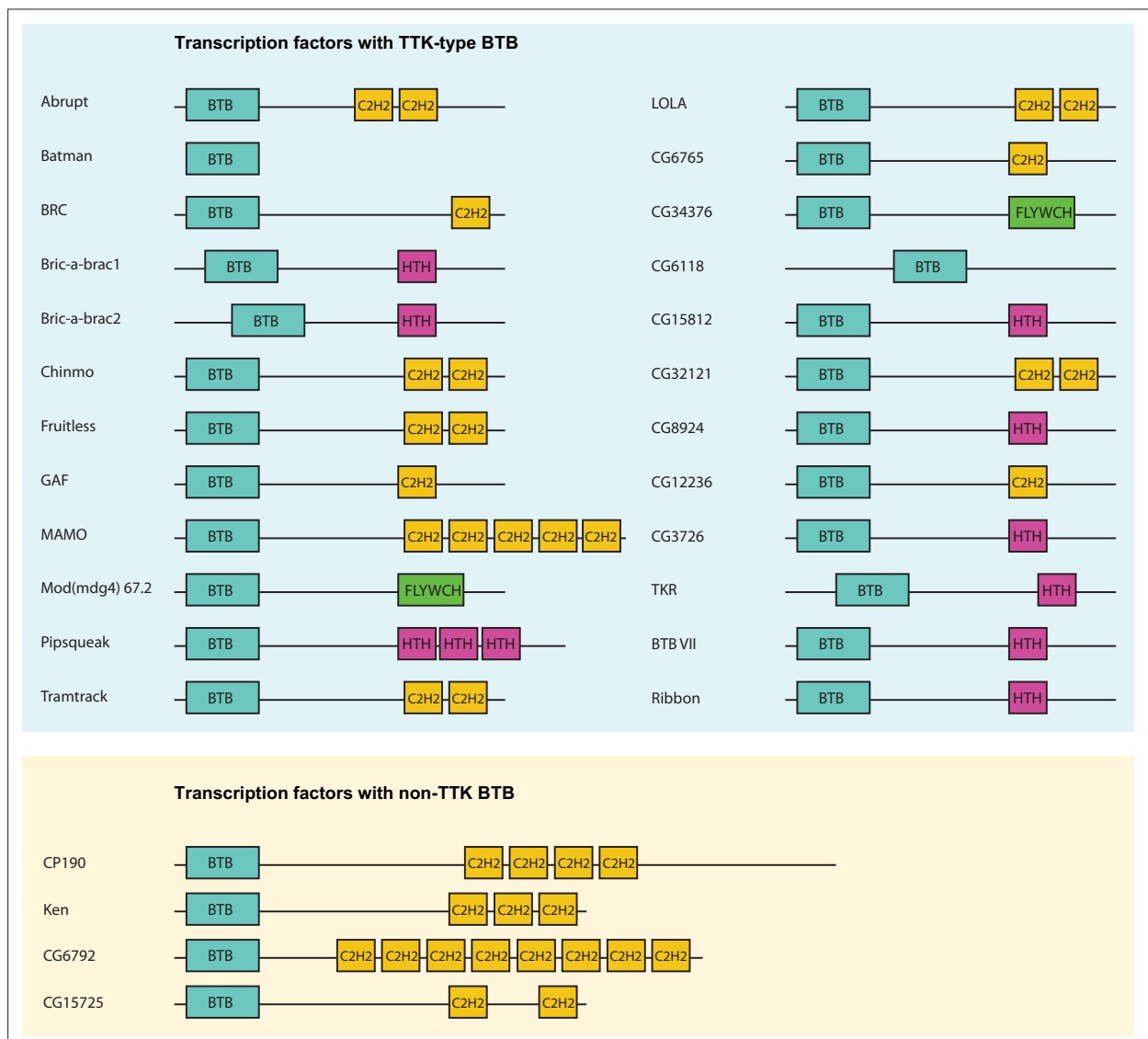

**Figure 1.** Schematic representation of *Drosophila* transcription factors with broad-complex, Tramtrack, and bric-a-brac (BTB) domains. Only one representative isoform is shown for each gene. Pipsqueak psq DNA-binding domain is depicted as 'HTH' since it is a type of helix-turn-helix (HTH) domain.

# Results

## Most BTB domains of transcription factors in *Drosophila melanogaster* are of TTK-type

The *Drosophila melanogaster* genome contains 28 genes encoding transcription factors with BTB domains (*Figure 1*). Many genes encode several BTB-containing protein isoforms, which differ in their C-terminal sequences. At two exceptional loci, *mod(mdg4)* and *lola*, multiple isoforms are formed by *trans*-splicing, the mechanism of which is still unclear (*Tikhonov et al., 2018*). The *mod(mdg4)* locus encodes at least 30 isoforms, most of which have FLYWCH domains (*Bradley and Andrew, 2001*). Seventeen of the 20 isoforms produced by the *lola* locus contain different C-terminal C2H2 zinc-fingers (*Horiuchi et al., 2003*).

BTB domains of transcription factors belong to the distinct subtype of BTB domains (ZBTB) with N-terminal extension playing a role in their homodimerization (*Bonchuk et al., 2023*). A characteristic feature of the TTK-type BTB domain is the presence of the FxLRWN motif at this N-terminal extension. Comparison of the amino acid sequence of BTBs from 28 *Drosophila* BTB-containing transcription factors showed that 24 BTB domains contain the characteristic 'TTK motif' (*Figure 2a and b*). The TTK-type BTB domains are usually located at the N-termini of proteins, however, in two proteins (CG6118, bric-a-brac2), the BTB domains are located in the middle of the protein (*Figure 1*). As an exception, Batman consists of only the BTB domain. Four transcription factors (CP190, CG6792, CG15725, and Ken) have BTB domains without a TTK motif (*Figure 1*).

## The TTK-type BTB domains display a broad heteromeric interaction propensity

The characteristic feature of the TTK-type BTB domains is their ability to selectively interact with each other (*Bonchuk et al., 2011*). We tested the interaction of all TTK-type BTB domains with BTB domains from Mod(mdg4), LOLA, Chinmo, CG8924, and GAF in a yeast two-hybrid assay (Y2H; *Figure 2c*, and *Supplementary file 2* and *Figure 2—figure supplements 1–6*). We found that all BTB domains can interact with themselves. 23 out of 24 TTK-type BTB domains also interacted with at least one of the GAF, LOLA, or Mod(mdg4) BTB domains. GAF, Mod(mdg4), LOLA, and CG8924 all interacted with at least 10 of the 24 TTK-type BTB domains, while Chinmo interacted with 18 TTK-type BTB domains (*Figure 2c*, *Supplementary file 2*). The homology between interacting domains was mostly in the range 35–47%, but could be as low as 25%. No obvious relationship between homology and hetero-meric interaction ability was found (*Supplementary file 3*).

We also tested the interaction between four non-TTK BTB domains (CP190, Ken, CG15275, and CG6792 (dPLZF)) and did not find heteromeric interaction between them (*Supplementary file 4*). The BTBs of CP190 and CG6792 formed homodimers in vitro, like all other classical BTB domains (*Figure 3—figure supplement 2*). The Ken and CG15725 BTB domains were insoluble after bacterial expression. Using Y2H, we detected only a small number of interactions between the non-TTK BTB domains of CP190 and Ken with the TTK-type BTB domains: CP190 interacted with GAF and Ribbon, while Ken interacted with Batman and Mamo (*Supplementary file 5* and *Figure 2—figure supplements 7–11*). These data also serve as an additional negative control for results showing the broad ability of TTK-type to heteromerization in Y2H assay. These results confirm that the TTK-type domains are functionally distinct from classical BTB domains.

## Cryo-electron microscopy reveals hexameric assembly of the BTB domain of CG6765 protein through a previously uncharacterized interface

To test the ability of TTK-type BTB domains to multimerize and shed light on their possible structure, we screened all TTK-type BTB domains from *Drosophila melanogaster* for the possibility of high-yield expression and soluble purification for subsequent structural analysis.

We expressed all 24 TTK-type *Drosophila* BTB domains as TEV-cleavable thioredoxin fusions. Out of 24 BTBs, only ten domains were soluble and stable after TEV-cleavage. Eight BTB domains appeared as a single peak on size-exclusion chromatography (SEC) with apparent Mw 90–150 kDa, Batman BTB eluted as a dimer (*Figure 3—figure supplements 3–4*), while CG32121 BTB formed

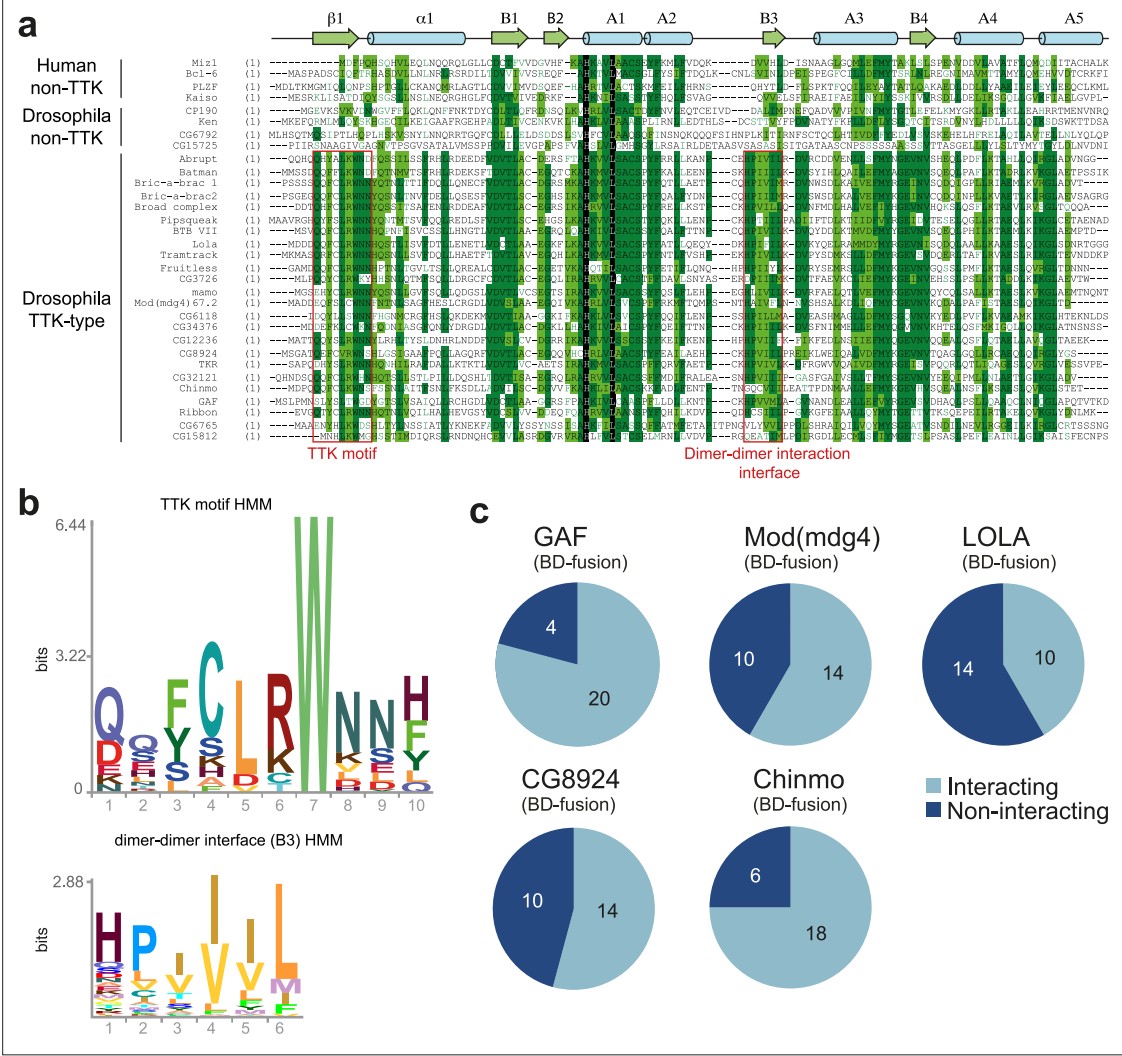

**Figure 2.** Characterization of the the Tramtrack group (TTK)-type broad-complex, Tramtrack, and bric-a-brac (BTB) domains. (**a**) Multiple sequence alignment of BTB domains from *Drosophila* transcription factors and a few human BTB domains with known 3D structures. Secondary structure elements are labeled according to *Stogios et al., 2005*.(**b**) Hidden Markov Model (HMM)-profile models for the TTK motif (upper) and the main dimer-dimer interaction interface (lower) were obtained for 14 Diptera species. (**c**) Testing of the GAF, Mod(mdg4), and LOLA BTB domains for interaction with all TTK-type BTB domains found in *Drosophila melanogaster*. Original data are shown in *Figure 2—figure supplement 1*.

The online version of this article includes the following figure supplement(s) for figure 2:

**Figure supplement 1.** Results of yeast two-hybrid assays.

**Figure supplement 2.** Results of yeast two-hybrid assays of interactions between the Tramtrack group (TTK) broad-complex, tramtrack, and bric-a-brac (BTB) domains (continued).

**Figure supplement 3.** Results of yeast two-hybrid assays of interactions between The Tramtrack group (TTK) Broad-complex, tramtrack, and bric-a-brac (BTB) domains (continued).

**Figure supplement 4.** Results of yeast two-hybrid assays of interactions between the Tramtrack group (TTK) broad-complex, tramtrack, and bric-a-brac (BTB) domains (continued).

**Figure supplement 5.** Results of yeast two-hybrid assays of interactions between the Tramtrack group (TTK) broad-complex, tramtrack, and bric-a-brac (BTB) domains (continued).

**Figure supplement 6.** Results of yeast two-hybrid assays of interactions between the Tramtrack group (TTK) broad-complex, tramtrack, and bric-a-brac (BTB) domains (continued).

**Figure supplement 7.** Results of yeast two-hybrid assays of interactions between the Tramtrack group (TTK) and non-TTK broad-complex, tramtrack, and bric-a-brac (BTB) domains.

*Figure 2 continued on next page*

*Figure 2 continued*

**Figure supplement 8.** Results of yeast two-hybrid assays of interactions between the Tramtrack group (TTK) and non-TTK broad-complex, tramtrack, and bric-a-brac (BTB) domains (continued).

**Figure supplement 9.** Results of yeast two-hybrid assays of interactions between the Tramtrack group (TTK) and non-TTK broad-complex, tramtrack, and bric-a-brac (BTB) domains (continued).

**Figure supplement 10.** Results of yeast two-hybrid assays of interactions between the Tramtrack group (TTK) and non-TTK broad-complex, tramtrack, and bric-a-brac (BTB) domains (continued).

**Figure supplement 11.** Results of yeast two-hybrid assays of interactions between the Tramtrack group (TTK) and non-TTK broad-complex, tramtrack, and bric-a-brac (BTB) domains (continued).

multiple peaks. Unfortunately, most of these BTBs had low solubility (~1 mg/mL) and, therefore, were not suitable for structural studies.

Only three BTB domains (CG6765$^{1-133}$, CG32121$^{1-147,}$ and LOLA$^{1-120}$) were soluble at concentrations above 5 mg/mL. CG32121$^{1-147}$ was excluded from further analysis due to its heterogeneous oligomeric state. Multiple crystallization trials led to nicely shaped crystals of CG6765$^{1-133}$ which, however, diffracted to above 6 Å, hampering the structure solution.

Hence, we used single-particle cryo-EM to elucidate the structure of BTB domain assemblies. To enhance particle contrast on cryo-EM images, we used the BTB domain of CG6765$^{1-133}$ fused to MBP (40 kDa), which resulted in a monomer Mw of 56 kDa. Iterative masked refinement excluding flexible MBP regions resulted in a final reconstruction of core CG6765$^{1-133}$ multimer at a resolution of 3.3 Å (full data processing flowchart is shown at *Figure 3—figure supplement 1*). The map clearly shows the hexameric assembly consisting of three dimers (*Figure 3a*). To model the atomic structure of CG6765 hexamers, we utilized an AlphaFold multimer implementation (*Evans et al., 2022*). The hexamer was predicted with high confidence and despite the loop regions the model fits well into the obtained cryo-EM map (*Figure 3b*). CG6765 hexamer consists of three canonical BTB dimers with extensive hydrophobic molecular contacts and main-chain hydrogen bonds (N78, V82, Y84, V86) forming a β-sheet between two parallel β4-strands (corresponding to B3 according to *Stogios et al., 2005*; *Figure 3b and c*). In addition, a bifurcated hydrogen bond is formed from the side chain hydroxy group of Y84 of one dimer to the sidechain of T77 and main-chains of either N78 or I76 of the second dimer. Furthermore, a few inter-dimer hydrogen bonds are formed between (i) T50 and T73, (ii) N78 and N80, and (iii) side chain hydroxy group of Y36 and P75/M70 main chain atoms. Interdimeric interactions were further strengthened by hydrophobic contacts between A2 (F66, M70, P75), the loop preceding B1 (I48), and B2/B1/B3 sheet (V34, V86) (*Figure 3b*), with the B3 strand formed by residues highly conserved only within the TTK group (*Figure 2a*). Residues with large hydrophobic or aromatic side-chains at positions corresponding to Y84 and V86 of CG6765 are characteristic for all TTK-type BTBs, whereas residues involved in stabilizing interactions at B1/A2 and adjacent loops are less conserved (*Figure 2a and b*). The N-terminal TTK-motif is located within the first β-strand and is involved only in intra-dimer stabilization forming antiparallel beta-sheet with B4 (*Figure 3b*). Central part of the hexamer forms a pore, residues of loop regions surrounding the pore are not conserved, thus it is unlikely that it can possess some physiological function.

While hexamerization of TTK-type BTB domains is unique for the ZBTB family, the formation of tetramers and pentamers is a common property of the T1/KCTD structural subclass of BTBs. However, the mechanism of multimerization is completely different (*Figure 3d*, *Figure 3—figure supplement 5*): T1/KCTD multimerize through the interaction between the B2 strand and A3 helix of adjacent subunits, further stabilized by contacts between the A4 helix and B3/A3 loop (*Dementieva et al., 2009*; *Ji et al., 2016*; *Kreusch et al., 1998*; *Minor et al., 2000*). The contribution of structural elements differs between families: the formation of T1 tetramers mostly relies on the B3/A3 loop interaction with the A3 helix, whereas the A3/A4 loop interaction with A4 is more important for the formation of KCTD pentamers (*Figure 3d*). Recently another subclass of multimer-forming human ZBTBs was described (*Mance et al., 2024*; *Park et al., 2024*). These domains use yet another interface to form large filamentous multimers (through A2/A4/A5 helices and the B3-A3/A4-A5 linkers of each dimer).

Thus, the structure of CG6765 BTB domain obtained with cryo-EM reveals the first hexameric structure of the BTB domain of the TTK group consisting of three canonical BTB dimers connected via a novel interface formed by two parallel β-strands which has not yet been implicated in BTB multimerization.

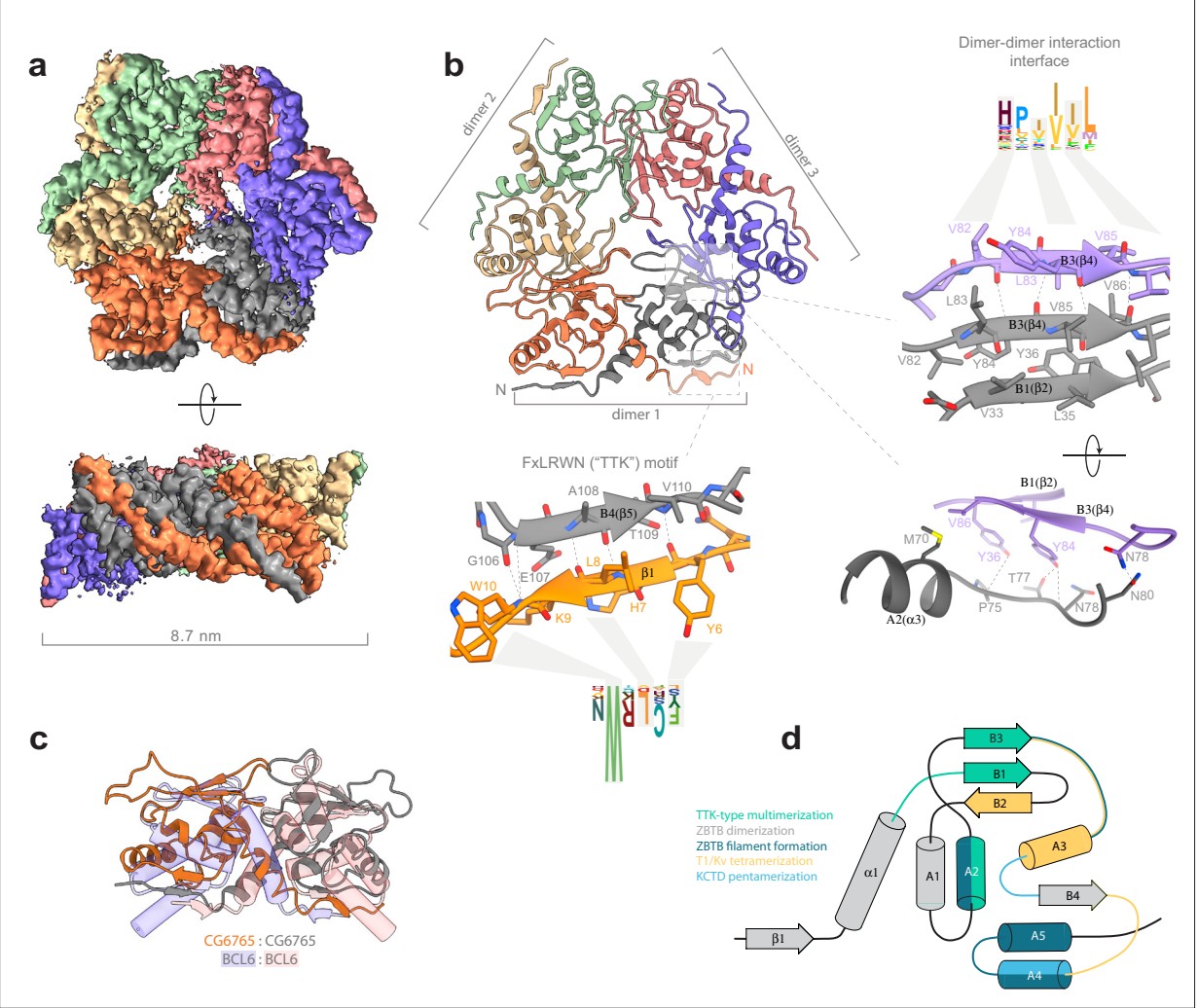

**Figure 3.** Cryo-EM structure of CG6765 broad-complex, tramtrack, and bric-a-brac (BTB) domain. (**a**) Cryo-EM map of CG6765 BTB domain. Map regions are colored according to corresponding protein chains. (**b**) Refined model of CG6765 BTB domain hexamer. Individual dimers are depicted, and details of dimer-dimer interaction interface and beta-sheet formation by FxLRWN ('the Tramtrack group, TTK') motif are shown at the right. Secondary structure elements are depicted according to **Stogios et al., 2005**. (**c**) Overlay of the dimeric subunit of CG6765 BTB domain and classical dimer of Bcl6 BTB domain (PDB ID: 1R28) (**Ahmad et al., 2003**). (**d**) Summary of the involvement of the secondary structural elements of BTB domains in multimerization. Overlays of BTB multimeric assemblies are shown at the **Figure 3—figure supplement 5**. (**d**) has been adapted from Figure 3A from **Bonchuk et al., 2023**.

The online version of this article includes the following figure supplement(s) for figure 3:

**Figure supplement 1.** Flowchart illustrating cryo-EM data processing of MBP-fused CG6765[1-133].

**Figure supplement 2.** Superdex S200 size-exclusion chromatography of the CP190 and CG6792 broad-complex, tramtrack, and bric-a-brac (BTB) domains.

**Figure supplement 3.** Superdex S200 size-exclusion chromatography of the Tramtrack group (TTK)-type broad-complex, tramtrack, and bric-a-brac (BTB) domains from *Drosophila*.

**Figure supplement 4.** Superdex S200 size-exclusion chromatography of the Tramtrack group (TTK)-type broad-complex, tramtrack, and bric-a-brac (BTB) domains from *Drosophila* (continued).

**Figure supplement 5.** Overlays of multimeric assemblies of CG6765 hexamer with.

## Hexamers are the main oligomeric state of TTK-type BTB domains in solution

As it was noted earlier, 8 out of 10 soluble BTB domains of the TTK group formed stable high-order multimers. The SEC profile for the representative BTB domain of LOLA had a single symmetric peak,

the position of which remained unchanged even upon 10-fold dilution (*Figure 4a*). Since the apparent Mw of 113 kDa determined relative to protein standards did not allow us to unequivocally determine the oligomeric state of LOLA (the predicted monomer mass is 15.25 kDa), it was analyzed using SEC-MALS, which provides the absolute Mw. The Mw determined by this method was 86.3 kDa, which was 5.7 times larger than the predicted monomeric Mw, and therefore was closest to a hexameric species. The monodispersity of the sample (Mw/Mn = 1.000) and the unchanged position on the elution profile upon dilution together indicated that the observed oligomer is stable. Consistent with this, we observed that MBP-fused LOLA also had a single symmetrical peak on the SEC profile, and its MALS-derived Mw of 315 kDa suggested a monodisperse (Mw/Mn = 1.000) ~5.6 mer (*Figure 4b*).

Since particles of BTB domains of CG6765 and LOLA are monodisperse in solution, we expected that BTB domain of LOLA would have a similar structure as was revealed by cryo-EM for CG6765. On the other hand, B3 element involved in hexamer formation is conserved among all TTK-type BTBs, but B3 of CG6765 substantially differs from others (*Figure 2a*). To further verify the structures and stability of stoichiometry of hexameric assemblies for LOLA BTB domain devoid of additional tags, we applied the SAXS method and AlphaFold modeling using CG6765 BTB as a reference. SAXS-derived structural parameters are listed in *Table 1*. Estimated molecular weights for LOLA and CG6765 BTB domains roughly correspond to hexamers and are in agreement with SEC-MALS and cryo-EM data. Then, we critically assessed AlphaFold models with different stoichiometries (from dimers to octamers) using the approximation of experimental SAXS data with curves calculated from the models using CRYSOL (*Svergun et al., 1995 Figure 4—figure supplement 1a, b*). For both CG6765 and LOLA BTBs, the theoretical scattering for hexameric models agreed best with the experimental data ($\chi^2$ values 1.7 and 3.6, respectively *Figure 4c and d*), while the fits from alternative oligomeric assemblies predicted by AlphaFold were inadequate (*Figure 4—figure supplement 1a, b*). The models of 7-mers contained largely disconnected hexamers and an additional subunit and were not considered. The models of 5-mers provided second-best fit to the experimental SAXS profiles, but were disregarded due to symmetry considerations, the contradiction to cryo-EM data, and also because they appeared to represent hexamers lacking one of the subunits. We believe that, while the presence of incomplete assemblies in vitro cannot be excluded, they do not represent the main oligomeric state of TTK-type BTB domains. Thus, SAXS data support that the prevalent oligomer state in the solution of LOLA and CG6765 BTB domains is the hexamer, in accordance with the CryoEM data. The overall structures of both hexamers are highly similar (*Figure 4c and d*).

We also modeled hexamer assemblies of different BTB domains of the TTK group (*Figure 4—figure supplement 2*) including Mod(mdg4) protein (*Figure 4—figure supplement 2b*), which is the best-studied protein of the family so far (*Golovnin et al., 2007*; *Melnikova et al., 2017*). All models had an architecture similar to those of the LOLA and CG6765 hexamers with interdimeric interface predicted with high confidence (*Figure 4—figure supplement 2a*). The dimer-dimer interaction interface is also conserved in Batman, which did not form multimers larger than dimer, according to SEC data. We, therefore, modeled a Batman BTB dimer and compared its structure with a LOLA dimer within the hexamer. We found no steric hindrance obstructing hexamer formation in Batman, although some hexamer-stabilizing interactions were absent in this case, for example, F42, which stabilized the LOLA hexamer, is T42 in Batman, in the cryo-EM structure of the CG6765 BTB this interaction is supported by Y36 from adjacent beta-strand (*Figure 4—figure supplement 2a and c*). From this observation, we suggest that the core β-sheet might be not sufficient for stable hexamer formation, and further stabilizing contacts are required, which may determine the specificity of the interaction. The dimer-dimer interaction interface was substantially different in the model of Chinmo BTB, which can still form multimers, as well as in Ribbon and CG15812. Unfortunately, we could not test the multimerization of the latter two domains due to strong aggregation; however, the AlphaFold model suggested that the β-strand involved in the dimer-dimer interaction remained unchanged, but the confidence of prediction was substantially lower (*Figure 4—figure supplement 2a and c*).

It is likely that unusually broad interactions between different TTK-type BTB domains (heteromeric interactions) occur due to the association of different homodimers through the interdimeric interfaces described above, which are highly similar in different members of this family (*Figure 2a*). Notably, the sequence of B3 strand of CG6765 is the least conserved among all TTK-type BTBs, that correlates with its lower ability for heteromeric interactions (*Figure 2—figure supplement 1*). We developed a set of substitutions of conserved hydrophobic residues involved in the inter-subunit β-sheet interface. We

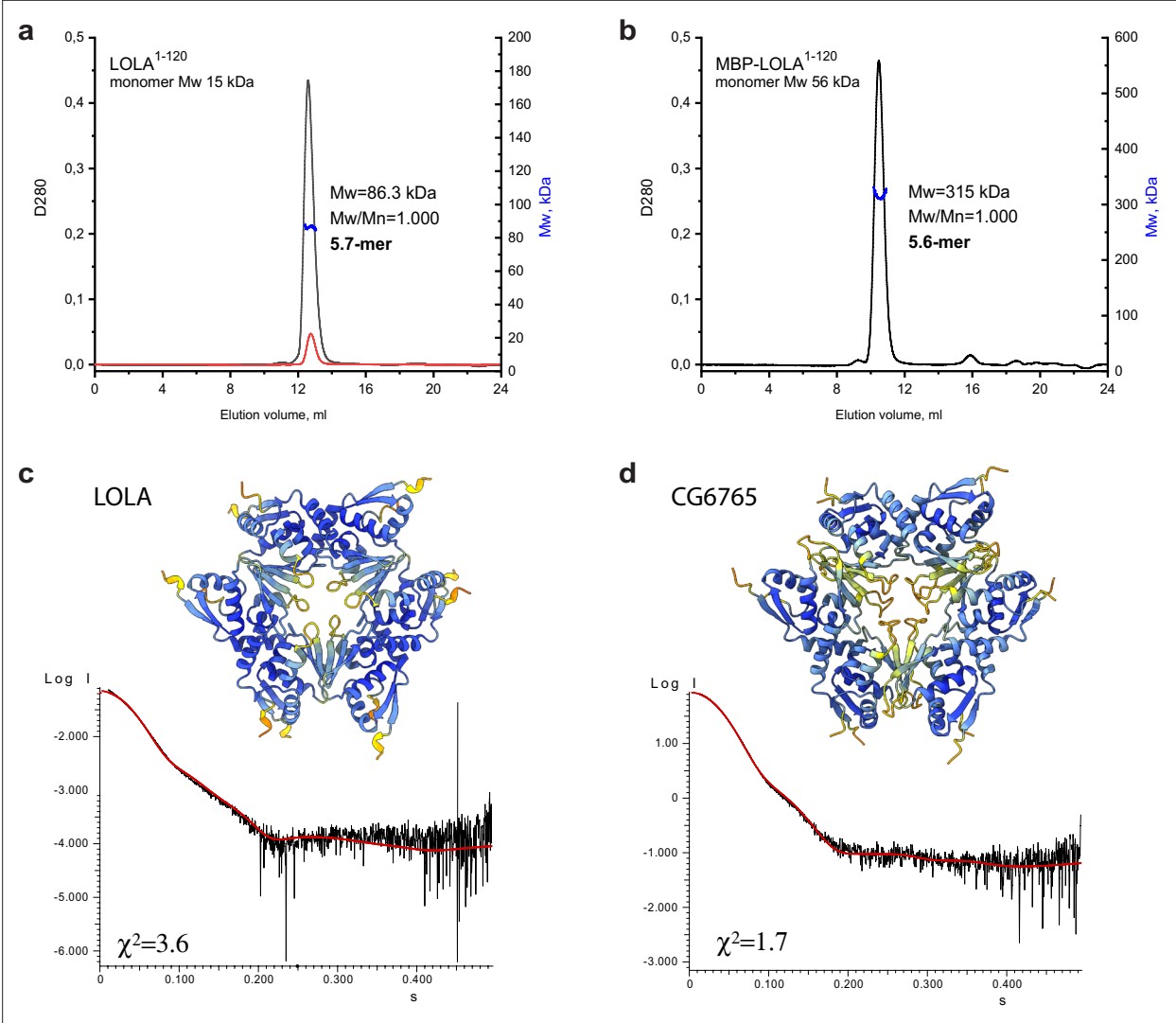

**Figure 4.** An integrative biology approach reveals the hexameric assembly of the Tramtrack group (TTK)-type broad-complex, tramtrack, and bric-a-brac (BTB) domains. Size-exclusion chromatography with multi-angle light scattering (SEC-MALS) data for LOLA. (**a**) and MBP-LOLA (**b**) showing the chromatographic peaks with the Mw distributions across each peak. Average Mw values in kDa, polydispersity index (Mw/Mn), and the formally calculated oligomeric state are shown. D280 designates optical density at 280 nm (absorbance units). The second Y axes are Mw, and kDa. Note that the 10-fold dilution of LOLA (black and red curves in panel **a**) did not cause any shift of the chromatographic peak, indicating the stability of the observed oligomer. AlphaFold2-derived models of LOLA (**c**) and CG6765 (**d**) oligomers and fits of their theoretical scattering data to the experimental SAXS data. Models are colored according to AlphaFold pLDDT values.

The online version of this article includes the following source data and figure supplement(s) for figure 4:

**Figure supplement 1.** Studying the possibility of different stoichiometry of BTB domain assemblies using SAXS.

**Figure supplement 2.** Molecular modeling of the Tramtrack group (TTK)-type broad-complex, tramtrack, and bric-a-brac (BTB) domains with AlphaFold.

**Figure supplement 3.** Testing the impact of single amino-acid substitutions in dimer-dimer interaction interface on the oligomerization status of the Tramtrack group (TTK)-type broad-complex, tramtrack, and bric-a-brac (BTB) domains.

**Figure supplement 4.** Superdex S200 size-exclusion chromatography of Thioredoxin-tagged LOLA broad-complex, tramtrack, and bric-a-brac (BTB) domains bearing mutations in dimer-dimer interaction interface.

**Figure supplement 4—source data 1.** Uncropped gel images.

**Figure supplement 5.** Superdex S200 size-exclusion chromatography of Thioredoxin-tagged Mod(mdg4) broad-complex, tramtrack, and bric-a-brac (BTB) domains bearing mutations in dimer-dimer interaction interface.

**Figure supplement 5—source data 1.** Uncropped gel images.

*Figure 4 continued on next page*

*Figure 4 continued*

**Figure supplement 6.** Superdex S200 size-exclusion chromatography of Thioredoxin-tagged CG6765 broad-complex, tramtrack, and bric-a-brac (BTB) domains bearing mutations in dimer-dimer interaction interface.

**Figure supplement 6—source data 1.** Uncropped gel images.

**Figure supplement 7.** Results of yeast two-hybrid assays of the impact of point mutations at the dimer-dimer interaction interface.

employed several strategies, first, to break inter-subunit main-chain H-bonds we mutated to prolines conserved hydrophobic residues with side-chains contacting another subunit (I70/I72 in LOLA, I71/F73 in mod(mdg4), and Y84/V86 in CG6765), second, we substituted these residues for lysine, which has large charged side chain to create electrostatic repulsion and make the surface more polar, and at last we applied conventional substitutions to alanine. Both single substitutions and substitutions of both residues at once were created. In general, all mutations led to a strong decrease in protein solubility and the appearance of aggregates in the SEC profiles (*Figure 4—figure supplements 3–6*), and often to an inability for self-interaction in Y2H assay (*Figure 4—figure supplement 7*) suggesting a general impact on protein folding. Only a small fraction of hexamers was present, along with the appearance of a fraction of dimers of comparable intensity, indicating the effect of the tested mutations on the formation of hexamers. A large peak of proteolytic fragments was also visible, suggesting protein misfolding and degradation (see *Figure 4—figure supplements 4–6*). Strong impact of mutations on protein stability much likely is a result of exposing large hydrophobic surfaces formed by A2 and B1/B2 with surrounding loops indicating their importance for stabilization of the hexamer. Mutational data support the importance of the integrity of dimer-dimer interaction interface for proper folding and stability of TTK group BTB domains. Such a strong impact of dimer-dimer interface mutations on protein folding and self-association does not allow to confidently study the effect of these mutations on heteromeric interactions. To get further insight into possible mechanism of heteromerization we used AlphaFold-Multimer to predict the structures of possible heteromer assemblies since it was found to be reliable in predicting hexameric assemblies of TTK-type BTB domains. Prediction results suggested that usage of both inter-dimer and conventional dimerization interfaces for heteromeric interactions are possible in various cases, with preference for one over another in different pairs (*Supplementary file 6*). Thus, most likely the presence of two potential heteromerization interfaces expands the heteromerization capability of these domains.

In summary, various approaches independently confirm that hexameric assembly of three dimers is the main oligomeric state of TTK group BTB domains.

## TTK-type BTB domains are specific to Arthropoda

In *D. melanogaster*, most transcription factor BTB domains are of TTK-type. We, therefore, investigated when these domains emerged over the course of evolution and how widespread in animals they are. We built the Hidden Markov Model (HMM) profile 'TTK motif' based on sequence alignment of TTK-type BTBs from 14 Dipteran species (*Figure 2b*) and performed a search within the main groups of Arthropoda and several other Metazoan species (including basal groups such as Onychophora, Tardigrada, and Nematoda). TTK-type BTB domains were not found outside of Arthropoda. The most basal clades in which they emerged were Crustaceans and Arachnoidea (*Figure 5a*). This agrees with a recent study which traced the origin of GAF and Mod(mdg4) insulator proteins to ancestral groups

**Table 1.** SAXS-derived structural parameters for CG6765 and LOLA broad-complex, tramtrack, and bric-a-brac (BTB) domains. Rg is the radius of gyration, Dmax – the maximum dimension of the particles, and Vp – is Porod volume – volume of the particles. The sample of CG6765[1-133] with the concentration of 1.5 mg/ml provided sufficiently high-quality scattering data, which were used for all fitting experiments, whereas data obtained for LOLA[1-120] at concentrations 1.0 and 3.0 mg/ml were merged to obtain the necessary quality. Samples with higher concentrations exhibited signs of aggregation and were excluded from analysis.

| Polypeptide | Sample concentration, mg/ml | Rg, nm | Dmax, nm | Vp, nm3 | Estimated molecular weight, kDa | Molecular weight of the monomer, kDa |
|---|---|---|---|---|---|---|
| CG6765[1-133] | 1.5 | 3.7 | 12.7 | 166 | 90.5–116.5 | 15.3 |
| LOLA[1-120] | merged data for 1.0 and 3.0 mg/ml | 4.1 | 14.5 | 189 | 108–136 | 15.2 |

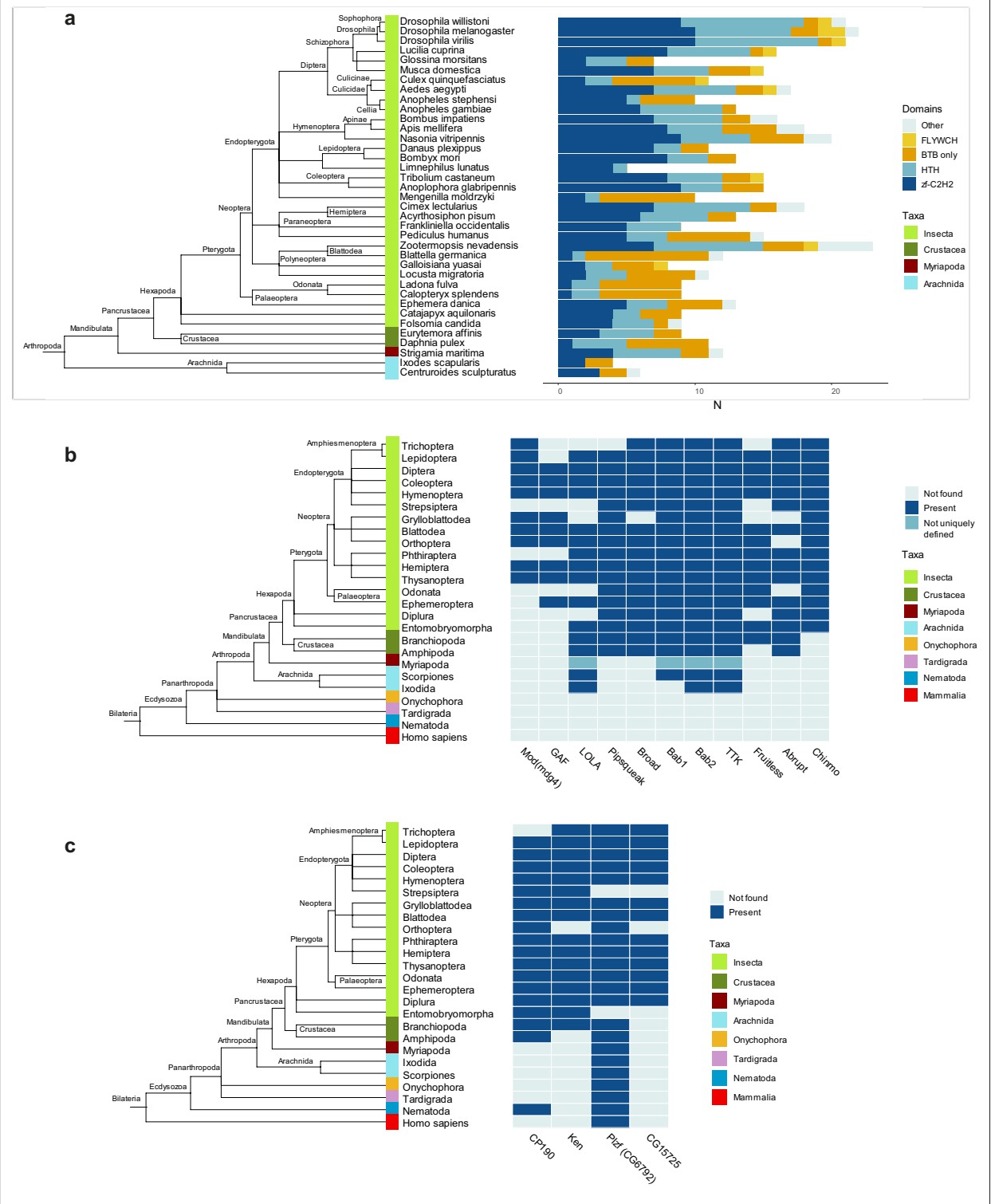

**Figure 5.** Domain architectures and orthologs of proteins with broad-complex, tramtrack, and bric-a-brac (BTB) domain of the Tramtrack group (TTK)-type in various Arthropoda lineages. (**a**) Phylogenetic analysis of the distribution of the main DNA and protein interaction domain types in TTK-type BTB proteins in proteomes of representatives of Hexapoda, Crustacea, Myriapoda, and Arachnida. Each type of BTB protein domain architecture is shown as a bar segment. Total search results are shown in *Supplementary file 7*. The orthologs of several *Drosophila* BTB domains of TTK-type (**b**) and non-TTK-type (**c**) in proteomes of key taxa in major Arthropod and other Ecdysozoa phylogenetic groups. The phylogenetic relationships among taxa are according to the NCBI Taxonomy Database. Azure blue – the ortholog is absent in the taxa, dark blue – the ortholog is present in the taxa, dark cyan – orthologs are not uniquely defined.

*Figure 5 continued on next page*

*Figure 5 continued*

The online version of this article includes the following figure supplement(s) for figure 5:

**Figure supplement 1.** Multiple sequence alignment of β1-B3 sequences of broad-complex, tramtrack, and bric-a-brac (BTB) domains (KLHL family from basal Metazoans and ZBTB family from Protostomia clades beyond Arthropoda) bearing N-terminal beta-strand according to AlphaFold-multimer predictions. Amino acid residues are colored with ClustalX colors (legend is shown below).

**Figure supplement 2.** AlphaFold2 modelling of non-Arthropoda TTK-type BTB domains.

of insects related to modern *Protura* and *Plecopthera* (**Pauli et al., 2016**). TTK-type BTB domains underwent a lineage-specific expansion in modern phylogenetic groups of insects, to the most extent in Diptera and Hymenoptera. In these species, TTK-type BTB domains almost completely replace transcription factors with classical dimeric BTB domains, as in *D. melanogaster* (**Figure 5a**). An ortholog search of the 11 best-studied *Drosophila* TTK-type BTB proteins revealed that the oldest proteins with TTK-type BTBs are LOLA, bric-a-brac2, and Tramtrack, for which orthologs were found in the most basal clades (**Figure 5b**). The same search for four non-TTK *Drosophila* BTB proteins revealed that CP190 and CG6792 (a *Drosophila* PLZF homolog) are the oldest: a CP190 ortholog was found in Nematodes and CG6792 orthologs are present in almost all other Metazoan taxa (**Figure 5c**). Notably, each non-TTK *Drosophila* BTB-C2H2 protein has only one isoform, whereas many factors with a TTK-type BTB have multiple isoforms, which typically differ in their DNA-binding domain, resulting in a wide diversity of these factors.

Most transcription factors with ZBTB also contain C2H2 or HTH DNA-binding domains. With the exception of Arachnida, in all examined Arthropoda species, the FLYWCH and HTH domains are found only in combination with the TTK-type BTB domains (**Figure 6a**). In the human proteome, BTB-containing transcription factors almost exclusively utilize C2H2 zinc-fingers as DNA-binding domains. Interestingly, TTK-type BTBs are usually associated with one or two C2H2 domains (only MAMO has five C2H2 domains), while mammalian transcription factors commonly have BTB domains in combination with arrays consisting of an average of five C2H2 domains (**Figure 6b**).

To trace the origin and evolution of N-terminal extension of ZBTB we searched Uniprot database for the proteins with the simultaneous presence of BTB and Zinc Fingers and run AlphaFold-Multimer predictions of dimeric assemblies of all BTB domains in proteomes of basal Metazoan taxa (*Amphimedon queenslandica*, sponge the most ancient metazoans (**Srivastava et al., 2010**), *Trichoplax adhaerens*, primitive metazoan organism (**Srivastava et al., 2008**) diverged over 600 million years ago), in proteome of *Monosiga brevicollis*, a member of Choanoflagellates, considered to be the closest metazoans ancestors (**King et al., 2008**), and in the proteomes of *Arabidopsis thaliana* and *Dictyostelium discoideum* as of members of sister to metazoan eukaryotic lineages. Cullin3-interacting domains KLHL-BTBs have similar to ZBTB architecture and dimerization surface, unlike ZBTBs they also possess additional 3-box/BACK alpha-helical extension serving as Cullin3-interacting domain (**Ji and Privé, 2013**). KLHL-BTBs are present in all eukaryotes and likely are predecessors of ZBTB domains (**Bonchuk et al., 2023**). According to AlphaFold modeling of dimers, all KLHL-BTB domains of plants and basal metazoans have α1 helix, but most of these domains from basal metazoans do not possess additional N-terminal beta-strand (β1) characteristics for ZBTB domains. We found only one KLHL-BTB (Uniprot ID: AA9VCT1_MONBE) with such N-terminal extension in Choanoflagellate proteome, one in *Dictyostelium* proteome (Q54F31_DICDI), and 7 (out of 43 BTB domains in total) and 13 (out of 81) such domains in *Trichoplax* and *Amphimedon* proteomes correspondingly (**Figure 5—figure supplement 1**). There was no significant sequence similarity of β1 element at the level of primary sequence (**Figure 5—figure supplement 1**). However, most of these domains bear 3-box/BACK extension and represent typical KLHL-BTBs which are member of E3 ubiquitin-ligase complexes, they are often associated with protein-protein interacting MATH domain or WD40 repeats. We found only one protein in *Trichoplax* proteome (B3RQ74_TRIAD) with β1 strand in BTB domain which is also devoid of 3-box/BACK, resembling ZBTB topology. Thus, likely emergence of BTB domains of this subtype occurred early in Metazoan evolution (**Figure 6c**). At this point, ZBTBs were not yet associated with zinc-fingers. According to our survey, the actual fusion of the ZBTB domain with zinc-finger domains occurred in the evolution of earlier bilaterian organisms since proteins with such domain architecture are not found in Radiata but are present in basal Protostomia and Deuterostomia clades (**Bonchuk and Georgiev, 2024**). AlphaFold prediction of possible hexameric assemblies of ZBTBs associated with zinc fingers from proteomes of organisms from several Protostomia clades (*Octopus*

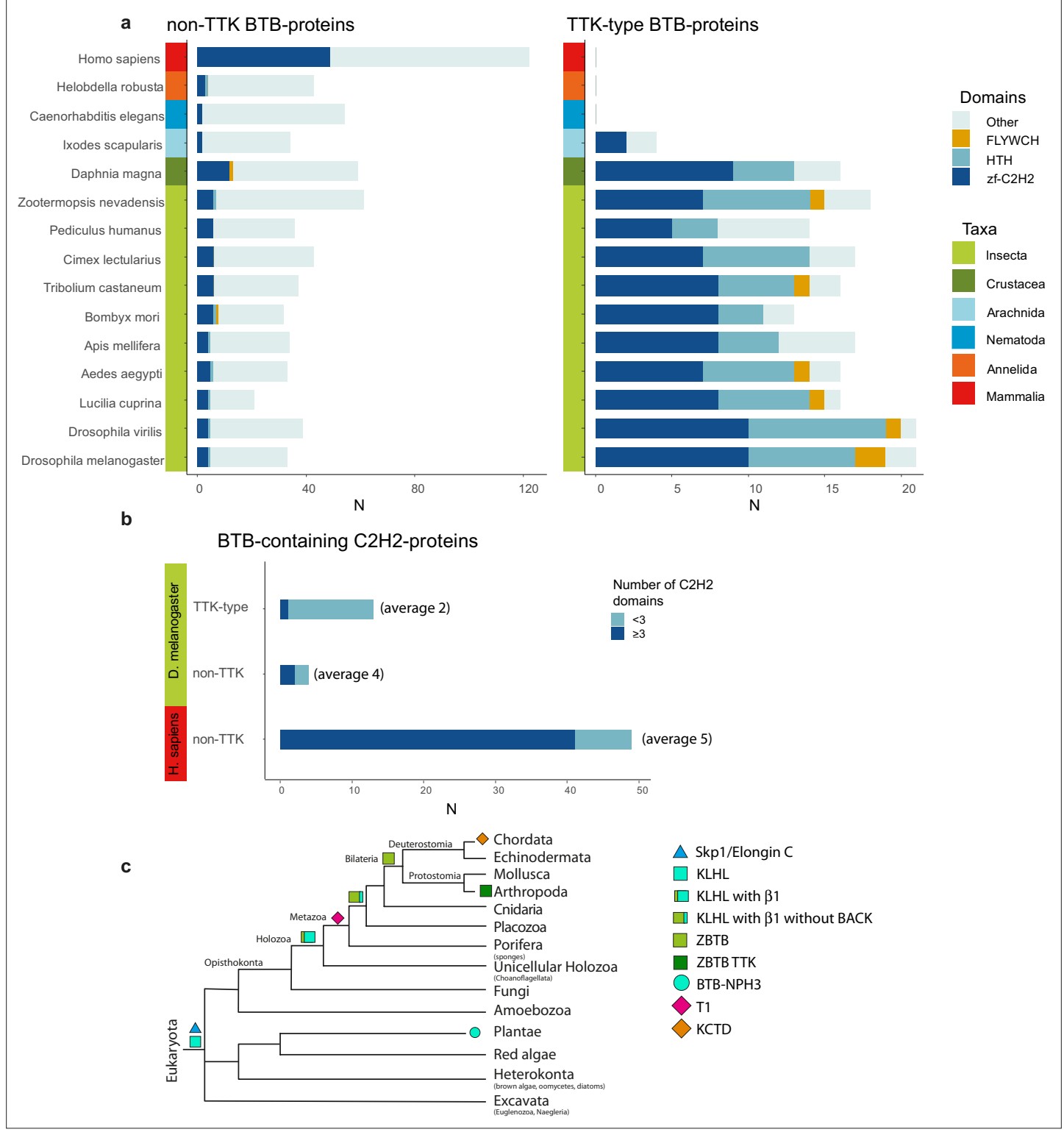

**Figure 6.** The Tramtrack group (TTK)-type broad-complex, tramtrack, and bric-a-brac (BTB) domains are specific to Arthropodan transcription factors. (**a**) Distribution of major DNA interaction (C2H2, HTH, FLYWCH) and other domain types in non-TTK (left) and TTK-containing (right) BTB proteins in the proteomes of 16 Metazoan species. Each type of BTB-associated protein domain is denoted as a bar segment. Figure represents proteins with all BTB subtypes, domains of the ZBTB subtype are almost exclusively associated with one of the DNA-binding domains. Total search results are shown in *Supplementary file 7* (TTK BTB domains) and *Supplementary file 8* (non-TTK BTB domains).(**b**) Representation of non-TTK and TTK-containing BTB proteins with less than three (dark cyan) and three or more (dark blue) C2H2 domains in *Drosophila melanogaster* and *Homo sapiens* proteomes. (**c**) Origin of different subtypes of BTB domains over the course of evolution. (**c**) has been adapted from Figure 6C from *Bonchuk et al., 2023*.

*bimaculoides, Aplysia californica, Lingula anatina, Caenorhabditis elegans, Capitella teleta*) did not yield any reliable models. None of β1 and B3 element sequences are similar to those of TTK-type BTB domains (*Figure 2*, *Figure 5—figure supplement 1*). We further extended our search across whole InterPro database using HMM profile built on the sequences involved in the hexamer formation ranging from A2 to B3 structural elements derived from *Drosophila* TTK-type BTB domains (*Supplementary file 9*). Arthropoda proteins were excluded and top-score hits were subjected to AlphaFold-Multimer modeling of the hexamer interface. Surprisingly, in some plant species we found a few BTB domains able to form TTK-like hexamers, close examination of sequence homology revealed that these proteins contain TTK-motif and are close relatives of Arthropodan BTBs and likely originated from lateral gene transfer (*Figure 5—figure supplement 2*). Other top-score domains were not predicted to form hexamers.

Taken together, our results suggest that proteins with TTK-type BTBs comprise a distinct group of transcription factors specific to Arthropoda.

## Discussion

The TTK-type BTB domains form a distinct group of ZBTB domains specific to Arthropodan transcription factors. Here, we studied the mechanism of multimer formation of TTK-type BTB domains using an integrative structural biology approach. Cryo-EM, SAXS, molecular modeling, and MALS revealed that these domains form hexamers consisting of three dimers assembled through a novel dimer-dimer interaction interface. Hence the mechanism of multimer assembly is completely different from that observed in multimeric BTB domains from the T1/KCTD family (*Figure 3d Ji et al., 2016*; *Long et al., 2007*). A neighbor surface (based on the B1-strand) was recently found to be involved in the Miz-1 BTB interaction with the HUWE-1 protein (*Orth et al., 2021*) and along with B3 was earlier implicated in Miz-1 tetramer formation in crystal (*Stead et al., 2007*). Apparently, this interface can be widely used for BTB-domain-mediated protein-protein interactions. Notably, the dimer-dimer interaction in Miz-1 involves conformational flexibility in this region, and monomers within each dimer are non-equivalent, with only one containing the B3-strand. Our structure and models show that dimers of TTK-type BTB domains consist of identical monomers; however, further structural studies will be required to elucidate precise details of this interface. Single amino acid substitutions further confirmed that residues at the dimer-dimer interface are critical for multimer formation. The characteristic conserved N-terminal sequence FxLRWN forms the first β-strand and was found to be important for specific dimer formation rather than for higher-order oligomerization as previously suggested (*Bonchuk et al., 2011*). This indicates that the FxLRWN motif at the N-terminus co-evolved together with the multimerization motif of the interface forming residues between the dimers. Probably FxLRWN is involved in binding to the yet unknown interaction partner. TTK-type BTB domains possess an unusually wide potential for heteromeric interactions despite a rather low sequence similarity. Heteromultimerization likely may occur through heterodimer formation as well as through the interaction of different dimers, since the dimer-dimer interface is highly similar in different TTK-type BTB domains. The presence of two potential heteromerization interfaces most likely expands the ability of these domains to form hetero-multimers. This is a unique property of this class of ZBTB domains, as most classical ZBTB domains form almost exclusively homodimers. Recently discovered subclass of filament-forming human ZBTBs (*Mance et al., 2024*; *Park et al., 2024*) evolved to use different interface to assemble in multimers. The convergent evolution of multimerization of ZBTB domains in different phylogenetic groups emphasizes the importance of multimer formation in some of their functions.

The expansion of the TTK-type BTB domains in insects suggests an important but poorly-understood functional role for their ability to form homo- and hetero-multimers. Most of the transcription factors of the TTK group bind to short degenerate DNA motifs. Thus, the multimerization of proteins via the BTB domain can increase the binding affinity for several motifs located in close vicinity. Such a mechanism was previously demonstrated for the pioneer and bookmarking protein GAF that is stably associated with chromatin (*Bellec et al., 2022*; *Gaskill et al., 2021*). GAF has only one zinc finger domain that binds to a GAGAG motif. It was shown that BTB oligomerization mediates strong co-operative binding of GAF to multiple sites but inhibits binding to a single motif (*Espinás et al., 1999*; *Katsani et al., 1999*) and promotes chromatin loop formation (*Li et al., 2023*).

Heteromultimerization of BTB domains can allow the formation of complexes of several TTK-type proteins on chromatin. The most striking example is the Batman protein, which consists only of the

BTB domain (**Mishra et al., 2003**). Batman forms complexes with Pipsqueak and GAF by interacting with their BTBs (**Faucheux et al., 2003**). Batman also interacts with the BAB1 BTB domain, and these proteins cooperate in the control of sex combs on male tarsa (**Gibert et al., 2007**). Since the Batman BTB can be recruited to chromatin only through interaction with other DNA-binding BTB proteins, it does not form homomultimers that would prevent it from interaction with other BTBs. Thus, TTK-type BTB transcription factors may constitute a regulatory network controlling common target genes.

In this study, we have shown that TTK-type BTB domains are Arthropod-specific and underwent lineage-specific expansion in modern insects. The *Drosophila* proteome contains 24 transcription factors with TTK-type BTB domains. Moreover, non-TTK-type BTB domains are found in only four *Drosophila* transcription factors, suggesting that the TTK-type BTB domain confers some evolutionary-important benefits. The most basal clades in which TTK-type BTBs can be traced are Crustacea and Arachnoidea. The only such proteins present in both taxa are Tramtrack orthologs, suggesting that it is the oldest member of the family.

Tracing the origin of ZBTB domains using bioinformatic and protein modeling approaches revealed that Cullin3-interacting KLHL BTBs likely are predecessors of ZBTB, evolving of the ZBTB-type N-terminal extension containing β1 element occurred early in metazoan evolution, however, the actual fusion of ZBTB with zinc-fingers in domain architectures happened over the course of evolution of Bilaterian organisms.

Another multimeric class of BTB domains, the T1 tetramerization domains of voltage-gated potassium channels, emerged also in metazoans. The first BTB domains of this type are found in sponges but not in Choanoflagellates (**Srivastava et al., 2010**). T1 BTBs are associated with the emergence of multicellularity and the need for cell-cell communication (**Srivastava et al., 2010**). These BTBs evolved distinct mechanisms of tetramer formation (**Figure 2d Bonchuk et al., 2023**). T1 BTBs further specialized in vertebrates, leading to the divergence of the KCTD subclass with various functions (**Liu et al., 2013**).

In humans and mouse, BTB-C2H2 proteins are encoded by at least 49 genes that are important regulators of development and commonly function as sequence-specific repressors of gene expression. In contrast to Arthropoda TTK-type BTB proteins, mammalian BTB-C2H2 proteins usually have many C2H2 domains that recognize extended and specific DNA motifs. Despite the fact that mammals have more highly-specific DNA-binding proteins, many of the TTK-type BTB transcription factors in Arthropods have multiple isoforms with different DNA-binding or protein interaction domains which, along with the ability to heteromultimerize, can result in a large diversity of complexes possessing a broad range of DNA-binding specificities.

In conclusion, TTK-type BTB proteins form a structurally and functionally distinct group of Arthropod key regulatory factors with unique functions in the process of cell differentiation and transcriptional regulation. What functional role is played by the ability of these transcription factors to form various combinations of heterologous complexes remains an open and very interesting question.

## Materials and methods

**Key resources table**

| Reagent type (species) or resource | Designation | Source or reference | Identifiers | Additional information |
|---|---|---|---|---|
| Gene (*Drosophila melanogaster*) | CG6765 | GenBank | NM_139976.2 | |
| Gene (*Drosophila melanogaster*) | Lola | GenBank | NM_170623.6 | |
| Strain, strain background (*Escherichia coli*) | BL21(DE3) | Novagen | 69450 | |
| Recombinant DNA reagent | pMALX(A) (plasmid) | **Moon et al., 2010** | | |
| Recombinant DNA reagent | pET32a(+) (plasmid) | Novagen | 69015 | |
| Recombinant DNA reagent | pGAD424 | TaKaRa bio | NCBI gi: 464015 | |
| Recombinant DNA reagent | pGBT9 | TaKaRa bio | NCBI gi: 470667 | |
| Software, algorithm | RELION 4.0 and 5.0beta | **Scheres, 2012 Kimanius et al., 2023** | | |
| Software, algorithm | CryoSPARC v4.3.1 | **Punjani et al., 2017** | | |
| Software, algorithm | PyEM | **Asarnow et al., 2019** | | |

*Continued on next page*

*Continued*

| Reagent type (species) or resource | Designation | Source or reference | Identifiers | Additional information |
|---|---|---|---|---|
| Software, algorithm | UCSF Chimera | *Pettersen et al., 2004* | | |
| Software, algorithm | Phenix.refine | *Liebschner et al., 2019* | | |
| Software, algorithm | ISOLDE | *Croll, 2018* | | |
| Software, algorithm | COOT | *Emsley et al., 2010* | | |
| Software, algorithm | Topaz | *Bepler et al., 2019* | | |
| Software, algorithm | ChimeraX v.1.6 | *Pettersen et al., 2021* | | |
| Software, algorithm | ATSAS package | *Franke et al., 2017* | | |
| Software, algorithm | AlphaFold2 | *Evans et al., 2022* | | |
| Software, algorithm | AlphaPulldown | *Yu et al., 2023* | | |
| Software, algorithm | HMMSEARCH | *Potter et al., 2018* | | |
| Software, algorithm | MEME | *Bailey et al., 2009* | | |
| Software, algorithm | MUSCLE | *Edgar, 2004* | | |
| Software, algorithm | biomaRt | *Durinck et al., 2009* | | |
| Software, algorithm | OrthoDB | *Kriventseva et al., 2019* | | |
| Software, algorithm | MEGA-X | *Kumar et al., 2018* | | |
| Software, algorithm | taxize | *Chamberlain and Szöcs, 2013* | | |
| Software, algorithm | CDD/SPARCLE NCBI | *Lu et al., 2020* | | |

## Bioinformatics

The search for orthologs for 24 *D. melanogaster* proteins with BTB domains containing ttk sequences was carried out in Metazoa, with the exception of Chordata, using the OrthoDB database (*Kriventseva et al., 2019*). There were no records for Mod(mdg4), GAF, and LOLA in OrthoDB, and orthologs of Fruitless were found only in Diptera. In order to fill up the data obtained from OrthoDB, orthologs for these proteins were searched in Ensembl using biomaRt (*Durinck et al., 2009*). The total number of identified orthologs was 3027. Amino acid sequences of orthologs of 23 ttk proteins of *D. melanogaster* in 14 Diptera species (five *Drosophila* species, two *Aedes* species, two *Anopheles* species, one *Culex* and fore flies) were aligned in MUSCLE (*Edgar, 2004*). The motif of the ttk domain in the resulting alignment was isolated using the EM algorithm by positional weight matrix (PWM) (at no limit motif E-value threshold and motif size 6–10 aa (*Figure 2a*)) in MEME (MEME SUITE) (*Bailey et al., 2009*), then in FIMO (MEME SUITE) we searched for the motif in the obtained database of orthologous sequences for all Metazoa (the threshold FDR value was 0.001). In addition, for the training set for 14 Diptera species, a hmm profile for the ttk domain was generated in MEGA-X (*Kumar et al., 2018*), and it was searched using HMMSEARCH (HmmerWeb version 2.41.1 *Potter et al., 2018*) in the UniProtKB database at E-value=10. Data on full protein sequences (n=259, excluding isoforms) deposited in UniProtKB were combined with previously obtained ttk-containing sequences of orthologs from OrthoDB and Ensembl (n=2208). In addition, in the large taxa closest to Arthropoda (several phyla of Protostomes: Tardigrada, Onychophora, Priapulida, Kinorhyncha, Loricifera, Nematoda, Annelida, Mollusca), the search for the ttk sequence motif in blastp and hmm-profile in HMMSEARCH were carried out; in the representatives of the above taxa, no proteins containing the ttk motif were found in non-redundant DBs.

Additionally, using the mentioned training set for 14 species of Diptera, a hmm-profile of the expanded dimer-dimer interface (including the 22 preceding amino acids) was built in HMMER 3.3.2 (*Finn et al., 2011*). The hmm-profile search was carried out among all proteins containing the BTB domain (303,255 sequences in total) retrieved from the InterPro database and including all kingdoms of living organisms in HMMER 3.3.2. The number of hits exceeding the $1\times10^{-5}$ E-value threshold was 129,357.

Phylogenetic classification was reconstructed in accordance with the NCBI Taxonomy Database using taxize (*Chamberlain and Szöcs, 2013*), and visualization of the phylogenetic tree is implemented

in ggtree (*Yu, 2020*). In the most basal clades (Crustacea, Arachnida, Myriapoda), the presence of the corresponding orthologs was predicted manually in blastp, due to the low similarity of amino acid sequences. We considered as orthologs, in addition to the previously annotated ones, only those proteins for which homology was observed in the structure, in both the BTB domain (including the presence of the ttk sequence) and the DNA binding domain. The presence/absence of some orthologs in individual orders of insects was also checked in blastp and HMMSEARCH (HmmerWeb version 2.41.1). In addition, the previously published data by *Pauli et al., 2016* were used for Mod (mdg4) and GAF.

The domain structure of the detected orthologs of ttk-containing proteins was studied in 37 reference species - representatives of 16 Hexapoda orders, two orders of Crustaceans and Arachnids, and one - Myriapoda (*Figure 5a*). The search for domains in the orthologous sequences was performed using the Pfam database in Batch CD-Search (CDD/SPARCLE NCBI [*Lu et al., 2020*]), the threshold E-value was taken to be 0.5. The detection results were filtered: incomplete and overlapping domains were removed. In sum, after filtering, the total number of domains found was 1055; the analyzed domains belonged to 26 PFAM families.

For 16 species with well-annotated proteomes belonging to the phyla Arthropoda, Nematoda, and Annelida, as well as for *Homo sapiens*, the sequences of all proteins containing BTB domains in their proteomes were searched using InterProScan in the Pfam protein domain database by the generalized hmm -BTB domain profile. In the set of the found sequences, the search for the ttk-domain motif in FIMO was carried out as described above, the ttk-containing proteins were filtered, and the remaining sequences were analyzed in Batch CD-Search in order to identify their domain structure. In total, 3028 domains belonging to 99 PFAM families were found in the sequences of BTB proteins without the ttk motif in 16 species.

Proteome-scale analysis of the simultaneous presence of C2H2 zinc-fingers and BTB domain and extraction of BTB domain sequences were carried out with an in-house built set of Python scripts available at https://github.com/errinaceus/cluster-counting, (copy archived at *errinaceus, 2024*).

AlphaFold predictions were run locally using AlphaFold 2.3 installation (*Evans et al., 2022*; *Jumper et al., 2021*) with a full set of genetic databases. Multiple predictions were arranged using AlphaPull-down wrapper scripts (*Yu et al., 2023*).

## Plasmids and cloning

cDNAs of BTB domains were PCR-amplified using corresponding primers (*Supplementary file 10*) and cloned into modified pET32a(+) vector (Novagen) encoding TEV protease cleavage site after 6X His-tag and Thioredoxin, and into pGAD424 and pGBT9 vectors (Clontech) in frame with GAL4 Activation or DNA-binding domains, respectively. PCR-directed mutagenesis was used to create constructs expressing mutant BTBs using mutagenic primers (*Supplementary file 10*). For MBP fusions cDNAs of BTB domains were cloned into pMALX(A) vector (*Moon et al., 2010*).

## Protein expression and purification

BL21(DE3) cells transformed with a construct expressing BTB domain fused with TEV-cleavable 6xHis-Thioredoxin were grown in 1 L of LB media to a D600 of 1.0 at 37 °C and then induced with 1 mM IPTG at 18 °C overnight. Cells were disrupted by sonication in buffer A (30 mM HEPES (pH 7.5), 400 mM NaCl, 5 mM β-mercaptoethanol, 5% glycerol, 0.1% NP40, 10 mM imidazole) containing 1 mM PMSF and Calbiochem Complete Protease Inhibitor Cocktail VII (1 μL/ml). After centrifugation, lysate was applied to a Ni-NTA column, and after washing with buffer B (30 mM HEPES (pH 7.5), 400 mM NaCl, 5 mM β-mercaptoethanol, 30 mM imidazole) was eluted with 300 mM imidazole. For cleavage of the 6X-His-thioredoxin-tag, 6X-His-tagged TEV protease was added at a molar ratio of 1:50 directly to the eluted protein, and the mixture was incubated for 2 hr at room temperature, then dialyzed against buffer A without NP-40 and applied to a Ni-NTA column. Flow-through was collected; dialyzed against 20 mM Tris-HCl (pH 7.4), 50 mM NaCl, and 1 mM DTT; and then applied to a SOURCE15Q 4.6/100 column (GE Healthcare) and eluted with a 50–500 mM linear gradient of NaCl. Fractions containing protein were concentrated, frozen in liquid nitrogen, and stored at –70 °C. Size-exclusion chromatography was performed in 20 mM Tris-HCl (pH 7.4), 200 mM NaCl, and 1 mM DTT using Superdex S200 10/300 GL column (GE Healthcare).

## SEC-MALS

Size-exclusion chromatography with multi-angle light scattering (SEC-MALS) detection was used to determine absolute Mw for LOLA and MBP-LOLA samples. Protein samples (1–5 mg/ml) were loaded individually on a Superdex 200 Increase 10/300 column (GE Healthcare), and the elution profiles were obtained using a tandem of sequentially connected UV-Vis Prostar 335 (Varian, Australia) and mini-DAWN detectors (Wyatt Technology, USA). The column was pre-equilibrated with filtered (0.1 µm) and degassed 20 mM Tris-HCl buffer, pH 7.6, containing 200 mM NaCl and 5 mM β-mercaptoethanol and was operated at a 0.8 ml/min flow rate. Data were processed in ASTRA 8.0 (Wyatt Technology, USA) using dn/dc equal to 0.185 and extinction coefficients ε(0.1%) at 280 nm equal to 0.80 ml/(mg cm) and 1.38 ml/(mg cm) for LOLA and MBP-LOLA, respectively. Additionally, apparent Mw values were determined from column calibration with standard proteins. Data were processed and presented using Origin 9.0 (Originlab, Northampton, MA, USA).

## Cryo-EM grid preparation and data collection

Quantifoil R1.2/1.3 Cu 300-mesh carbon grids were glow-discharged for 60 s at 20 mA using a GloQube (Quorum) instrument. 4 µl of the MBP-CG6765[1-133] sample (4 µg/µl) were applied to the freshly glow-discharged grids and plunge-frozen in LN2-cooled liquid ethane using a Vitrobot Mark IV (Thermo Fisher Scientific) with a blotting time of 2.5 s. Temperature and relative humidity were maintained at 4°C and 100%, respectively. Grids were clipped and loaded into a 300-kV Titan Krios G4i microscope (Thermo Fisher Scientific, USA) equipped with a Selectris X energy filter and a Falcon 4i (Thermo Fisher Scientific, USA) direct electron detector. Micrographs were recorded at a nominal magnification of 165,000X corresponding to a calibrated pixel size of 0.729 Å. A total of 9765 movies were recorded with a total dose of ~40 electron/Å2 per movie. The Defocus range was set between −0.5 µm and −2 µm. Data collection and processing statistics are summarized in *Supplementary file 1*.

## Cryo-EM data processing

*Figure 3—figure supplement 1* illustrates the data processing workflow for the MBP-CG6765[1-133] dataset. The following pre-processing steps were performed with cryoSPARC Live v4.3.1 (*Punjani et al., 2017*). Movie stacks were motion-corrected and dose-weighted, and contrast transfer function (CTF) estimates for the motion-corrected micrographs were calculated. Particles were initially picked with a blob-picker using subset-selected micrographs, and these were used for reference-free two-dimensional (2D) classification to generate picking templates. Subsequent image processing was carried out with cryoSPARC v4.3.1 (*Punjani et al., 2017*). The templates were used to train a picking model in Topaz (*Bepler et al., 2019*), which was subsequently used to pick particles from the whole dataset. Auto-picking using Topaz from 9765 micrographs yielded ~100,000 particles. Initial models were generated without imposing symmetry (**C1**) using stochastic gradient descent in cryoSPARC Live. Best initial model was subjected to homogeneous and heterogeneous refinement rounds and the best classes were used to create templates for another round of template picking, followed by 2D classification and training the new Topaz picking model, which was used for the final picking round. 480721 particles were picked. Particles were classified with three-dimensional (3D) heterogenous refinement using four classes, resulting in 197,562 particles.

To generate a high-resolution reconstruction, particles were re-extracted in 448 pixels box size followed by 3D refinement with a local angular search using the map from the previous processing in cryoSPARC as reference. A mask including only map fragments corresponding to the hexamer core was generated using UCSF Chimera (*Pettersen et al., 2004*) and RELION v.4.0 (*Scheres, 2012*). The mask was used in subsequent 3D refinement jobs and CTF refinements in CryoSPARC. The particles were exported to RELION using PyEM (*Asarnow et al., 2019*) and further processed in RELION 4.01. After 3D refinement followed by CTF refinement, Bayesian polishing was applied. 3D refinement on the polished particles, followed by CTF refinement and another round of 3D refinement with Blush regularization (in RELION 5.0beta *Kimanius et al., 2023*), yielded a reconstruction to ~3.3 Å overall resolution with C1 symmetry. We realized that Blush regularization did not improve the nominal resolution, however, it slightly improved the quality of the map especially in the core region of the complex. In the peripheral solvent exposed part, the map without blush regularization appeared to be of better quality and was, therefore, used to build the loop regions of the complex. Both maps were deposited in the EMDB.

AlphaFold model of the hexamer was used as the initial model, it was fitted into the reconstruction using UCSF Chimera (*Pettersen et al., 2004*), followed by manual real-space refinement in Coot, and further refined with Phenix.refine (*Liebschner et al., 2019*) and ISOLDE (*Croll, 2018*). Figures were prepared with UCSF ChimeraX (*Pettersen et al., 2021*).

## Yeast two-hybrid assay

Yeast two-hybrid assay was performed as described (*Bonchuk et al., 2011*).

## SAXS data collection and analysis

Synchrotron radiation X-ray scattering data were collected using standard procedures on the BM29 BioSAXS beamline at the ESRF (Grenoble, France) as described previously (*Bonchuk et al., 2020*). Data analysis was performed using the ATSAS software package (*Franke et al., 2017*). Approximation of the experimental scattering profiles using calculated scattering curves was performed with CRYSOL (*Svergun et al., 1995*). The molecular mass (MM) of the protein was calculated using several algorithms implemented in the ATSAS package (*Franke et al., 2017*).

## Acknowledgements

This work was supported by the Russian Science Foundation – project 19-74-10099-P to AB (expression and purification of proteins and their mutants), project 19-74-30026-P to PG (analysis of protein-protein interactions) and by Ministry of Science and Higher Education of the Russian Federation — grant 075-15-2019-1661 (structural and bioinformatic analysis). Funding for open access charge: Ministry of Science and Higher Education of the Russian Federation and Russian Science Foundation. The single-particle cryo-EM work was financially supported by the KAUST Baseline Grant BAS/1/1107-01-01. NNS and KMB acknowledges that SEC-MALS work was supported by the Ministry of Science and Higher Education of the Russian Federation. We are grateful to Dr. Alexander Kuklin (Joint Institute of Nuclear Research, Dubna) for help in SAXS data collection. We acknowledge the European Synchrotron Radiation Facility for provision of synchrotron radiation facilities and we would like to thank the staff of the ESRF for assistance and support in using beamline BM29. We thank Andrey Moiseenko (MSU, Moscow, Russia) for his help in the negative-staining EM data collection.

---

## Additional information

### Funding

| Funder | Grant reference number | Author |
| --- | --- | --- |
| Russian Science Foundation | 19-74-10099-P | Artem N Bonchuk<br>Konstantin I Balagurov<br>Konstantin M Boyko<br>Anna D Burtseva |
| Russian Science Foundation | 19-74-30026-P | Artem N Bonchuk<br>Konstantin I Balagurov<br>Anastasia M Khrustaleva<br>Olga V Arkova<br>Karina K Khalisova<br>Pavel G Georgiev |
| Ministry of Science and Higher Education of the Russian Federation | 075-15-2019-1661 | Artem N Bonchuk<br>Konstantin I Balagurov<br>Olga V Arkova |
| KAUST Baseline Grant | BAS/1/1107-01-01 | Rozbeh Baradaran<br>Andreas Naschberger |

The funders had no role in study design, data collection and interpretation, or the decision to submit the work for publication.

---

## Author contributions
Artem N Bonchuk, Conceptualization, Software, Formal analysis, Supervision, Funding acquisition, Investigation, Visualization, Methodology, Writing - original draft, Project administration; Konstantin I Balagurov, Rozbeh Baradaran, Konstantin M Boyko, Nikolai N Sluchanko, Olga V Arkova, Karina K Khalisova, Investigation; Anastasia M Khrustaleva, Formal analysis, Investigation; Anna D Burtseva, Formal analysis; Vladimir O Popov, Conceptualization; Andreas Naschberger, Conceptualization, Resources, Formal analysis, Supervision, Funding acquisition, Writing - review and editing; Pavel G Georgiev, Conceptualization, Supervision, Funding acquisition, Project administration, Writing - review and editing

## Author ORCIDs
Artem N Bonchuk ⓘ https://orcid.org/0000-0002-0948-0640
Konstantin I Balagurov ⓘ https://orcid.org/0000-0002-8650-1839
Konstantin M Boyko ⓘ https://orcid.org/0000-0001-8229-189X
Nikolai N Sluchanko ⓘ https://orcid.org/0000-0002-8608-1416
Anna D Burtseva ⓘ https://orcid.org/0000-0003-0675-1043
Olga V Arkova ⓘ https://orcid.org/0000-0002-5353-0748
Andreas Naschberger ⓘ http://orcid.org/0000-0002-7275-5459

## Decision letter and Author response
Decision letter https://doi.org/10.7554/eLife.96832.sa1
Author response https://doi.org/10.7554/eLife.96832.sa2

---

# Additional files

## Supplementary files
• MDAR checklist

• Supplementary file 1. Cryo-EM data collection and processing statistics.

• Supplementary file 2. Summary of the ability of GAF, Mod(mdg4), Chinmo, CG8924 and LOLA BTB domains to interact with other TTK-type BTB domains in yeast two-hybrid assay.

• Supplementary file 3. Pairwise level of similarity between GAF and Mod(mdg4) BTB domains with other TTK-type domains.

• Supplementary file 4. Testing of the interactions between the non-TTK BTB domains of C2H2 proteins.

• Supplementary file 5. The ability of non-ttk BTB domains of CP190, CG6792, CG15725 and Ken proteins to interact with other TTK-type BTB domains in yeast two-hybrid assay.

• Supplementary file 6. AlphaFold2-Multimer – modeled heteromeric interactions of TTK-group BTB domains with interaction confirmed in Y2H assay.

• Supplementary file 7. Total domain search results for proteins with TTK-type BTB domains.

• Supplementary file 8. Total domain search results for proteins with non-TTK-type BTB domains.

• Supplementary file 9. Total search results for HMM-profile built on consensus sequence of dimer-dimer interaction interface.

• Supplementary file 10. Oligonucleotides used for cloning.

## Data availability
The cryo-EM maps (blushed-regularized and normal regularization) and PDB model files have been deposited in the Protein Data Bank under the PDB entry code 8RC6 and in the EMDB with entry code EMD-19049. SAXS data have been deposited in the Small Angle Scattering Biological Data Bank (https://www.sasbdb.org) under accession codes SASDP59 (merged data for LOLA1-120 at 1.0 mg/ml and 3.0 mg/ml), SASDP49 (CG67651-133 at 1.5 mg/ml). Atomic models (both native and exactly corresponding to expression constructs) and reports of SAXS approximation are provided as supplementary files. Results of bioinformatic analysis are provided as supplementary tables.

The following datasets were generated:

| Author(s) | Year | Dataset title | Dataset URL | Database and Identifier |
|---|---|---|---|---|
| Bonchuk AN, Naschberger A, Baradaran R | 2024 | Cryo-EM structure of hexameric BTB domain of *Drosophila* CG6765 protein | https://www.rcsb.org/structure/8RC6 | RCSB Protein Data Bank, 8RC6 |
| Bonchuk AN, Naschberger A, Baradaran R | 2024 | Cryo-EM structure of hexameric BTB domain of *Drosophila* CG6765 protein | https://www.ebi.ac.uk/emdb/EMD-19049 | Electron Microscopy Database, EMD-19049 |
| Bonchuk A | 2024 | BTB domain of longitudinals lacking (LOLA) protein | https://www.sasbdb.org/data/SASDP59/ | Small Angle Scattering Biological Data Bank, SASDP59 |
| Bonchuk A | 2024 | BTB domain of CG6765 protein at 1.5 mg/ml | https://www.sasbdb.org/data/SASDP49/ | Small Angle Scattering Biological Data Bank, SASDP49 |

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
