## [Editor Report]

This important study investigates Tramtrack–like BTB domains of metazoan transcription factors using Cryo–EM microscopy, evolutionary and fold prediction analyses. The research presents compelling evidence for the structural basis of the multimerization and explores the evolutionary history of this family. This study will be of particular interest to structural and evolutionary biologists.

---

## [Decision Letter]

**Decision letter after peer review:**

Thank you for submitting your article "The Arthropoda–specific Tramtrack group BTB protein domains use previously unknown interface to form hexamers" for consideration by *eLife*. Your article has been reviewed by 2 peer reviewers, and the evaluation has been overseen by a Reviewing Editor and Volker Dötsch as the Senior Editor.

Essential revisions (for the authors):

1) Discuss the alternative multimer assembly interfaces, the role of various motifs and conserved residues.

2) Infer the evolutionary ancestors in the light of metazoan transcription factor evolution.

3) Investigate or interpret the effects of mutations on the formation of the BTB stable dimers.

*Reviewer #2 (Recommendations for the authors):*

Introduction:

– A few points in the Introduction would benefit from stating more clearly. It would be useful to state upfront the numbers of predicted human and *Drosophila* BTB proteins, and the numbers of these that are thought to be transcription factors. It would be relevant to name the other DNA–binding domains found in human BTB–TFs (i.e. to be consistent with the naming of other *Drosophila* DNA–binding domains). It should be clarified that the psq DNA–binding domain of Pipsqueak is a type of HTH domain (particularly as the DNA–binding domains of Pipspeak are simply depicted as "HTH" domains in Figure 1).

TTK–type BTB domains are specific to Arthropoda:

– Figure 5c is the wrong Figure (i.e. it does not correspond to the description referred to in the text or legend [rather it seems to be a version of Figure 6a]).

– Some parts would benefit from clarifying whether the discussion is around BTB proteins in general, or whether it relates specifically to BTB–domain transcription factors. The statement "Most BTB proteins also contain C2H2 or HTH DNA–binding domains" (start of second paragraph page 13) is somewhat ambiguous – this statement could relate to either the TTK–type BTB proteins or to the BTB–domain transcription factors (not all of which are of the TTK type), whereas many of the BTB proteins represented in Figure 6a are of the non–TTK type and do not represent transcription factors (hence do not have C2H2 or HTH domains).

– Please clarify whether TTK–type BTB domains are only found in transcription factors.

Hexameric organisation of the TTK–type BTB domains:

– The cryo–EM, SEC–MALs and SAXS data convincingly indicate a hexameric organisation of the CG6765 and LOLA TTK–type BTB domains. However, the contacts in the CG6765 cryo–EM structure need discussing in more detail. Although there is mention of contacts between A2 and the B1/B3 sheet, these need describing (and A2 needs labelling in the Figures). Although the authors state "..with the B3 strand formed by residues highly conserved only within the TTK group", this region in CG6765 appears significantly less well–conserved than in other TKK–type BTB domains (Figure 2a), and specific residues highlighted in Figure 3b do not in fact appear to be well–conserved.

– It is relevant that the authors were unable to produce stable BTB dimers by mutation of the B3 residues involved in the inter–subunit β–sheet interface involved in the hexameric assembly (such mutations led to protein aggregation and unfolding). This is surprising given that the mammalian transcription factor BTB domains form stable dimers with an "exposed" B3. This might imply that although B3–B3 interactions are critical for hexamer formation, other interactions unique to the hexamer (e.g. between A2 and B1/3) are also relevant (i.e. exposure of such regions following hexamer disruption might then result in aggregation/unfolding).

Heteromeric interactions between TTK–type BTB domains:

– Although the yeast two hybrid experiments demonstrate heteromeric interactions between various TTK–type BTB domains, there is no evidence that these are specific or that the B3 β–sheet interface is involved. This work would need validation using additional types of interaction assay.

– Please check that Figure S12 (yeast two–hybrid experiments carried out using mutant TTK–BTB domains) is referred to in the text.

---

## [Author Response]

Essential revisions (for the authors):Reviewer #2 (Recommendations for the authors):Introduction:– A few points in the Introduction would benefit from stating more clearly. It would be useful to state upfront the numbers of predicted human and *Drosophila* BTB proteins, and the numbers of these that are thought to be transcription factors. It would be relevant to name the other DNA–binding domains found in human BTB–TFs (i.e. to be consistent with the naming of other Drosophila DNA–binding domains). It should be clarified that the psq DNA–binding domain of Pipsqueak is a type of HTH domain (particularly as the DNA–binding domains of Pipspeak are simply depicted as "HTH" domains in Figure 1).

We made suggested corrections**.***TTK–type BTB domains are specific to Arthropoda:*

– Figure 5c is the wrong Figure (i.e. it does not correspond to the description referred to in the text or legend [rather it seems to be a version of Figure 6a]).

Thanks for your attention, occasionally we inserted older version of the figure. We replaced the figure with a correct one.

– Some parts would benefit from clarifying whether the discussion is around BTB proteins in general, or whether it relates specifically to BTB–domain transcription factors. The statement "Most BTB proteins also contain C2H2 or HTH DNA–binding domains" (start of second paragraph page 13) is somewhat ambiguous – this statement could relate to either the TTK–type BTB proteins or to the BTB–domain transcription factors (not all of which are of the TTK type), whereas many of the BTB proteins represented in Figure 6a are of the non–TTK type and do not represent transcription factors (hence do not have C2H2 or HTH domains).

Thanks, we added these clarifications and emphasized that BTBs of transcription factors belong to ZBTB class.

– Please clarify whether TTK–type BTB domains are only found in transcription factors.

We added this. All TTK-type BTB domains are transcription factors. The only formal exception is Batman protein which consists only of BTB domain and does not bind to DNA itself, however its interactions with a number of DNA-binding BTB proteins has been described which suggests its function in transcription regulation.

Hexameric organisation of the TTK–type BTB domains:– The cryo–EM, SEC–MALs and SAXS data convincingly indicate a hexameric organisation of the CG6765 and LOLA TTK–type BTB domains. However, the contacts in the CG6765 cryo–EM structure need discussing in more detail. Although there is mention of contacts between A2 and the B1/B3 sheet, these need describing (and A2 needs labelling in the Figures). Although the authors state "..with the B3 strand formed by residues highly conserved only within the TTK group", this region in CG6765 appears significantly less well–conserved than in other TKK–type BTB domains (Figure 2a), and specific residues highlighted in Figure 3b do not in fact appear to be well–conserved.

We improved description of inter-dimeric molecular interface

– It is relevant that the authors were unable to produce stable BTB dimers by mutation of the B3 residues involved in the inter–subunit β–sheet interface involved in the hexameric assembly (such mutations led to protein aggregation and unfolding). This is surprising given that the mammalian transcription factor BTB domains form stable dimers with an "exposed" B3. This might imply that although B3–B3 interactions are critical for hexamer formation, other interactions unique to the hexamer (e.g. between A2 and B1/3) are also relevant (i.e. exposure of such regions following hexamer disruption might then result in aggregation/unfolding).

We agree, strong impact of mutations onto protein stability much likely is a result of exposing large hydrophobic surface consisting of A2 and B1/B2 with surrounding loops indicating their importance for stabilization of hexamer. We added this to the description of effects of mutations.

Heteromeric interactions between TTK–type BTB domains:– Although the yeast two hybrid experiments demonstrate heteromeric interactions between various TTK–type BTB domains, there is no evidence that these are specific or that the B3 β–sheet interface is involved. This work would need validation using additional types of interaction assay.

We agree that our data do not explain the basis for heteromeric interactions. Design of mutations in B3 β-sheet proved to be complicated, using of biochemical methods to study the principles of heteromer assembly also does not seem to be feasible since most TTK-type BTBs tend to form aggregates and are difficult to be expressed and purified. But most important issue is that demonstrated ability of heteromer assembly through B3 in few tested pairs cannot be applied for all pairs, some of them may use different mechanism. We used AlphaFold to predict possible mechanisms of heteromer assemblies. AlphaFold suggested that usage of both B3 and conventional dimerization interfaces for heteromeric interactions are possible in various cases, with preference of one over another in different pairs. Thus, most likely the presence of two potential heteromerization interfaces extends the heteromerization capability of these domains. We changed the text accordingly.

– Please check that Figure S12 (yeast two–hybrid experiments carried out using mutant TTK–BTB domains) is referred to in the text.

The figure (Figure 4—figure supplement 7 in the new version) is referenced under section describing effects of mutations in B3.